# Immune Thrombocytopenic Purpura as a Hemorrhagic Versus Thrombotic Disease: An Updated Insight into Pathophysiological Mechanisms

**DOI:** 10.3390/medicina58020211

**Published:** 2022-02-01

**Authors:** Claudia Cristina Tărniceriu, Loredana Liliana Hurjui, Irina Daniela Florea, Ion Hurjui, Irina Gradinaru, Daniela Maria Tanase, Carmen Delianu, Anca Haisan, Ludmila Lozneanu

**Affiliations:** 1Department of Morpho-Functional Sciences I, Discipline of Anatomy, “Grigore T. Popa” University of Medicine and Pharmacy, Universității str 16, 700115 Iasi, Romania; claudia.tarniceriu@umfiasi.ro; 2Hematology Clinic, “Sf. Spiridon” County Clinical Emergency Hospital, 700111 Iasi, Romania; 3Department of Morpho-Functional Sciences II, Discipline of Physiology, Grigore T. Popa University of Medicine and Pharmacy, 700115 Iasi, Romania; 4Central Clinical Laboratory-Hematology Department, “Sf. Spiridon” County Clinical Emergency Hospital, 700111 Iasi, Romania; carmendelianu@gmail.com; 5Department of Morpho-Functional Sciences I, Discipline of Imunology, “Grigore T. Popa” University of Medicine and Pharmacy, Universității str 16, 700115 Iasi, Romania; 6Department of Morpho-Functional Sciences II, Discipline of Biophysics, “Grigore T. Popa” University of Medicine and Pharmacy, 700115 Iasi, Romania; ion.hurjui@umfiasi.ro; 7Department of Implantology Removable Dentures Technology, “Grigore T. Popa” University of Medicine and Pharmacy, Universității str 16, 700115 Iasi, Romania; irina.gradinaru@umfiasi.ro; 8Department of Internal Medicine, “Grigore T. Popa” University of Medicine and Pharmacy, 700111 Iasi, Romania; tanasedm@gmail.com; 9Department of Biochemistry, “Grigore T. Popa” University of Medicine and Pharmacy, 700115 Iasi, Romania; 10Surgery Department, “Grigore T. Popa” University of Medicine and Pharmacy, Universității str 16, 700115 Iasi, Romania; anca.haisan@umfiasi.ro; 11Emergency Department, “Sf. Spiridon” Emergency County Hospital, 700111 Iasi, Romania; 12Department of Morpho-Functional Sciences I, Discipline of Histology, “Grigore T. Popa” University of Medicine and Pharmacy, Universității str 16, 700115 Iasi, Romania; ludmila.lozneanu@umfiasi.ro; 13Department of Pathology, “Sf. Spiridon” Emergency County Hospital, 700111 Iasi, Romania

**Keywords:** immune thrombocytopenic purpura (ITP), thrombosis, hemorrhagic disease, regulatory cells, ITP treatment

## Abstract

Immune thrombocytopenic purpura (ITP) is a blood disorder characterized by a low platelet count of (less than 100 × 10^9^/L). ITP is an organ-specific autoimmune disease in which the platelets and their precursors become targets of a dysfunctional immune system. This interaction leads to a decrease in platelet number and, subsequently, to a bleeding disorder that can become clinically significant with hemorrhages in skin, on the mucous membrane, or even intracranial hemorrhagic events. If ITP was initially considered a hemorrhagic disease, more recent studies suggest that ITP has an increased risk of thrombosis. In this review, we provide current insights into the primary ITP physiopathology and their consequences, with special consideration on hemorrhagic and thrombotic events. The autoimmune response in ITP involves both the innate and adaptive immune systems, comprising both humoral and cell-mediated immune responses. Thrombosis in ITP is related to the pathophysiology of the disease (young hyperactive platelets, platelets microparticles, rebalanced hemostasis, complement activation, endothelial activation, antiphospholipid antibodies, and inhibition of natural anticoagulants), ITP treatment, and other comorbidities that altogether contribute to the occurrence of thrombosis. Physicians need to be vigilant in the early diagnosis of thrombotic events and then institute proper treatment (antiaggregant, anticoagulant) along with ITP-targeted therapy. In this review, we provide current insights into the primary ITP physiopathology and their consequences, with special consideration on hemorrhagic and thrombotic events. The accumulated evidence has identified multiple pathophysiological mechanisms with specific genetic predispositions, particularly associated with environmental conditions.

## 1. Introduction

Immune thrombocytopenic purpura (ITP) is a blood disorder characterized by a low platelet count of (less than 100 × 10^9^/L) [1,2]. ITP is associated with thrombocytopenia that has as clinical expression bleeding and hemorrhages in mucosa or skin [3,4]. ITP is a heterogeneous disease where clinical outcome and response to treatment display varied biologic behavior with a better outcome in the case of children and a worse outcome in adult cases [5,6].

According to the latest Thrombocytopenia International Working Group consensus, the incidence of ITP varies between children (1.9 and 6.4 per 100,000 per year) and adults (3.3 and 3.9 per 100,000 per year) [7,8]. Most ITPs manifest both in pre- and post-menopausal women and an increased incidence is reported in both sexes in the sixth decade of life [9].

International guidelines classify ITP into two major subtypes: primary and secondary ITP. In general, primary ITP is an acquired autoimmune disorder, characterized by platelet consumption through a high level of platelet destruction and/or development. These events are due to suppression of megakaryocytes and dysfunctional primary hemostasis through the collapse of immune tolerance mediated by cross-reactive anti-platelet autoantibodies [10,11]. 

Some of the major risk factors for secondary ITP include autoimmune disorders such as systemic lupus erythematosus, rheumatoid arthritis, Evans syndrome, Sjögren’s syndrome, and antiphospholipid syndrome [12,13]. The lymphoproliferative diseases (especially chronic lymphocytic leukemia), immunodeficiency (common variable immune deficiency), chronic infections due to the bacterial or viral proteins (human immunodeficiency virus (HIV), hepatitis C virus (HCV), Epstein-Barr virus (EBV), cytomegalovirus (CMV), *Helicobacter pylori*, and drugs are also reported to increase the risk of secondary ITP [14,15].

Currently, based on ITP outcomes, current guidelines are subdivided ITP into three phases, which can help to categorize the disease outcome: (i) First phase (newly ITP) which appears in the first 3 months from diagnosis); (ii) second phase (persistent ITP) which appears in 3 months–1 year from diagnosis); (iii) third phase (chronic ITP) which appears in more than 1 year from diagnosis [7,12]. 

Molecular (biological) and clinical studies indicate that children usually are considered newly diagnosed with spontaneous remission, while adult patients harbor refractory ITP forms [5,7,11]. Severe ITP should be considered a separate group that refers to the presence of severe bleeding without assuming an indolent course and this requires urgent and complex treatment [16]. All this evidence supports the idea that these phases are not homogenous and can have good or poor clinical outcomes.

Despite efforts to adapt a diagnostic protocol for ITP, the specific guidelines for diagnosis of ITP disease have not yet been achieved. Furthermore, there are no preventive measures for ITP and options for high-risk patients. Therefore, the gold standard for ITP diagnosis is difficult to find. Many cases of ITP can have unusual presentation. This includes other conditions that can lead to pseudothrombocytopenia and/or a lack of other pathognomonic evidence of ITP suspicion [17].

No laboratory analysis supports the diagnosis. A careful workup of a patient’s examination and correct patient clinical information is very useful. ITP can exhibit varied clinical features (behavior) that make definitive diagnosis challenging, as it can be mistaken for other blood disorders. Complete cell counting using the peripheral blood smear and examination of serum used for detection of characteristic platelet-specific autoantibodies may be utilized to facilitate the diagnosis, especially when the differentiation of the ITP is unclear or when the lesion has had a partial physical examination. Bone marrow examination is not a gold standard for ITP diagnosis but most of the time it is a requirement, taking into consideration that ITP is an exclusion diagnostic.

In addition, many other tests, recommended in only a few of the available guidelines, have been subsequently investigated including: (i) platelet surface glycoproteins (GP-specific) detected by antibodies tests; (ii) direct monoclonal antibody immobilization of platelet antigens (MAIPA); (iii) functional platelet assay [11,18]; (iv) mass cytometry [19]; (v) direct flowcytometric immunobead assays [20], and prove the diagnosis of ITP [21]. However, it seems that free platelet autoantibodies or autoantibodies bound to own platelets identified in patient serum express a high sensitivity and specificity approach, and it is commonly found during cases suspected of having ITP [22]. 

However, these antibodies can only be detected in 60% of the patients. In the remaining 40% of the cases, the antibodies are not present, but this is not a criterion for ITP exclusion, the explanation being that the pathophysiological mechanisms are complex and intricated.

Nowadays, several test methods have been developed to detect biomarkers such as the expression of CD61 (P-selectin) [19].

The goal of our paper is to review the various immunological mechanisms of ITP and its comprehensive cause of tendency to thrombosis although clinical reviews of thrombosis in ITP have appeared in recent years. A detailed review of the latest pathophysiology can help us to have a better approach of the disease.

## 2. ITP as a Hemorrhagic Disease

ITP is an organ-specific autoimmune disease in which the platelets and their precursors become targets of a dysfunctional immune system [9]. This interaction leads to a decrease in platelet number (thrombocytopenia) and, subsequently, to a bleeding disorder that can become clinically significant with hemorrhages in skin, on the mucous membrane, or even intracranial hemorrhagic events [23]. The incidence varies, having a slightly higher number in children than in adults [23]. 

The clinical manifestation of ITP varies between the patients [24]. Bleeding is the most common manifestation (bleeding of skin, oral cavity, or gastrointestinal tract). Purpura may appear without a precipitating event. Intracranial hemorrhage is the most serious and feared complication [24].

The basic mechanism of autoimmunity in ITP relies on a disequilibrium between effectors and regulatory cells [25,26,27,28,29], disequilibrium leading to increased platelet clearance at the same time with defective thrombopoiesis [23].

### Impact of Immune Response in ITP

During the 1950s, the involvement of the humoral-mediated immune response in the development of ITP was demonstrated [30] and the first antibodies to be detected as being involved in this process were IgGs [31]. Other isotypes such as IgA and IgM were also detected, even if only in association with IgG [32]. These antibodies targeted glycoproteins (GPs) or GP complexes (GPIIb/IIIa and GPIb/IX/V and less commonly GPIa/IIa, IV, or VI) [33]. These antibodies function as opsonins, coating the platelets and facilitating their endo-phagocytosis performed by macrophages carrying Fc receptors that bind to the Fc region of the coating antibodies. The receptors involved in this process are low-affinity Fcγ receptors, namely FcγRIIa and FcγRIIIa, which contain immune receptor tyrosine-based activation motifs (ITAMs) in the intracellular domain. Besides these activator receptors, there are FcγR receptors, namely FcγRIIb, that are inhibitory, because they carry an intracellular tyrosine-based inhibitory motif (ITIM). The consequence of this receptor ligation is the inhibition of phagocytosis and the release of pro-inflammatory cytokines by macrophages and dendritic cells. Therefore, the ratio between different FcγRs (activator and inhibitory) is important for the correct regulation of the humoral-mediated immune response [23].

The same antibodies (using the same Fc region) can induce the complement-mediated lysis of platelets through the classical pathway of complement activation. The process of complement activation occurs after the antibodies cover the surface of the platelets through binding with the above-mentioned targets [34,35]. GPIIb/IIIa and GPIb/IX are the main targets of autoantibodies with complement activation abilities [36]. These findings are emphasized by studies showing that the degree of platelet destruction is increased in the presence of the complement [37].

Researchers have described alternative mechanisms which are independent of the Fc region of the antibody. It was shown that ITP-autoantibodies can induce glycan modifications on platelet surface glycoproteins (GPs). Liver cells, expressing Ashwell–Morrell receptors are involved in the clearance of platelets carrying these surface GP changes [38]. Quach et al. suggested another Fc-independent mechanism. They showed that nonresponding ITP patients produce autoantibodies against the ligand-binding domain of GPIb/IX; the antibody binding to this domain results in the activation of GPIb/IX through crosslinking, with subsequent unfolding of its mechanosensory domain and platelet destruction [39]. Singh et al. showed that a subgroup of autoantibodies determined the cleavage of sialic acid on the surface of human platelets and megakaryocytes in ITP, this desialylation being responsible for the cell–extracellular matrix protein interaction impairment and, therefore, for the impaired platelet adhesion and megakaryocyte differentiation [40].

Other research groups showed that ITP autoantibodies are involved in platelet apoptosis, this fact being supported by data showing that depolarization of the mitochondrial transmembrane potential, bcl-2 family protein expression, as well as activation of caspase-3 and caspase-9 are involved in platelet apoptosis [41,42].

Besides the already known mechanisms responsible for platelet destruction, a mechanism resulting in platelet-impaired production and mediated by antibodies has been recently described [23]. The production of platelets is the result of a complex process, called megakaryopoiesis, that takes place in the bone marrow. This process involves molecular and cellular changes initiated and regulated by a growth factor named thrombopoietin (TPO), produced in the liver [43,44]. TPO is responsible for the differentiation of HSCs into megakaryocytes which finally release the platelets into the bloodstream [45]. Mesenchymal stem cells are cells that provide support for megakaryopoiesis by secreting cytokines such as IL-6, IL-10, Il-11, prostaglandins, stem cell factor (SCF), and leukemia-inhibiting factor [46]. Bone marrow-derived MSCs have been proven to have an abnormal morphology showing increased apoptosis and a decreased proliferation rate [47].

An explanation of the fact that the autoantibodies affect platelet formation by megakaryocytes could be that megakaryocytes, during differentiation, express the same GPIb and GPIIb/IIIa which are the targets of ITP autoantibodies found on platelets [23]. The autoantibody recognition of these targets leads to inhibition of megakaryocyte maturation and platelet formation [48,49]. Studies showed impaired megakaryocyte maturation and reduced platelet formation in experiments using HSC undergoing differentiation in the presence of ITP autoantibodies against GPIb/IX and GPIIb/IIIa [50,51]. Other research studies showed that anti-αvβ3 (a member of the integrin family of cell adhesion molecules) autoantibodies might have a selective inhibitory influence on megakaryocyte adhesion and migration during ITP pathogenesis [52]. The presence of autoantibodies against the TPO receptor (TPO-R) c-Mpl has also been observed [53].

The influence of B cells on the pathogenesis of ITP is not only related to the production of autoantibodies. There are studies proving that Breg cells are involved in ITP patients [54,55]. Bregs (CD19+ CD24hiCD38hi) secrete IL10 through which they recruit Tregs and reduce the function of Th cells; therefore, keeping peripheral tolerance. Functional dysfunction of Bregs was noticed in non-splenectomized ITP patients with chronic ITP [54].

Cell-mediated immunity has been proven to be involved in the non-antibody mechanisms of ITP as well [5]. It has been demonstrated that T cytotoxic cells (CTLs) from patients with ITP have a higher rate of proliferation and a lower rate of apoptosis, a fact that leads to a higher amount of IL-2, IFN-γ, and IL-10 secreted by them; this secretion being responsible for lower CD4+CD25+Foxp3+ Treg cell levels and efficiency in patients with active disease [56,57,58,59,60,61]. 

There are probably three main mechanisms through which T cells’ function dysregulation is involved in the pathophysiology of ITP [62]. One of them is the alteration of the ratio Th1/Th2 in favor of Th1. It has been shown that increased precursors of Th1 and a cytokine profile related to them (elevated amounts of IL-2 and IFNγ) coexist with chronic ITP [62], disfavoring the Th-2 cell number [63]. IL-10 in reduced amounts was identified in patients with active disease in comparison with those in remission or healthy controls [64]. In turn, elevated levels of IL-10 were identified in children with chronic ITP [64]. A second way through which T cells seem to be involved is the secretion of cytokines that interfere with megakaryocyte maturation and platelet release [62]. TGF-β levels were inversely correlated with the activity of the disease [65]. 

The low level of TGF-β in ITP patients leads to the production of a high level of autoantibodies against autologous platelets because the TGF-β level corresponds to a certain level of regulatory B and T cells that keep the autoimmune response under control [66]. Since TGFβ stimulates thrombopoietin release, which leads to increased platelet production, low TGFβ levels may represent the reason for a low platelet number in this disease [67].

In a third row, cell-mediated cytotoxic lysis of platelets by CTL might also be involved in the pathogenesis of ITP [67]. Type 1 helper T cells stimulate CTLs, macrophages, and the activity of NK cells as well as the production of IgG subclasses involved in antibody-dependent cell cytotoxicity. This profile of Th1 cells corresponds to that of Th cells in ITP patients [49]. The studies showing upregulation of interferon-γ, IL-2Rβ, perforin together with granzyme A, granzyme B, and Apo-1/Fas gene (the mediators of CTL induced apoptosis) are in favor of the T cell-mediated cytotoxicity relationship with platelet destruction in ITP [67].

Additional mechanisms through which T cells are involved in the pathogenesis of ITP are the increase in Th17 cells, decrease in T reg lymphocytes [9], and dysfunction of T follicular helper cells (TFH). ITP seems to be characterized through a high number of Th-17 cells [68,69], although other studies seem to confirm these findings [70,71]. These are proinflammatory cells that secrete high amounts of IL-17 and IL-22, and they differentiate from Th0 through another pathway which is then followed by Th1 and Th2 [26]. ITP is not the only autoimmune disease in which Th17 is involved [26]. IL-17 is increased during active disease in both children and adults [69] and IL-22 is reported to be elevated in the active phase of disease and decreased in patients responsive to dexamethasone [72]. Tregs, as mentioned above, form a subset of T cells with a CD4+CD25-^high^ Foxp3+ that play an important role in maintaining self-tolerance in physiological conditions through their ability to inhibit effector CD4+, CD8+ T cells, and B cells [73] and to induce DCs that are tolerogenic [74]. A reduced number of these cells [75] has been described in ITP patients with severe bleeding and thrombocytopenia as well as reduced immunosuppressive activity of Tregs from ITP patients in vitro [76]. The amount of IL-10 and IL-35, anti-inflammatory cytokines secreted by T regs, is reduced in ITP patients [77,78]. Studies have suggested that Treg-deficient mice develop thrombocytopenia mediated by IgG anti-platelet antibodies against GPIb/IX because the transfer of purified Tregs into Treg-deficient mice leads to the restoration of platelet count [79]. 

T follicular helper cells (TFH), as mentioned above, have a role in ITP pathogenesis because they are involved in B cell activation in secondary lymphoid organs followed by antibody production [67]. An expansion of TFH with a CD3+CD4+CXCR5+ICOS+PD-1+ profile in the germinal centers of the splenic follicles has been reported and their number is correlated with the germinal center B cells and the resulting plasma cells. TFH cells express CD40L and produce IL-21 and it was shown in vitro that, following stimulation with IL-21 and CD40, the B cells of ITP patients become plasma cells producing anti-GPIIb/IIIa antibodies and this indicates an important role of TFH cells in the pathogenesis of ITP [80].

Cells belonging to the innate immune response, through their mode of action, have an important role in the autoimmune response, such as that in ITP. Such cells, including DCs, macrophages together with B cell lymphocytes, have the role of scanning the environment, internalizing, processing, and presenting foreign antigens to lymphocytes through proteins belonging to the major histocompatibility complex [81,82]. DCs are the most efficient APCs [83] and different studies have shown their impairment in ITP [83]. In ITP, dendritic cells can phagocyte apoptotic platelets and subsequently stimulate naïve-specific T cells. Even though the phagocytic abilities of DC are similar in ITP and healthy controls, the expression of co-stimulatory molecules CD80 [84] and CD86 is higher, and they produce larger amounts of IL-12 than do controls [85]. Besides antigen presentation, TLR-mediated recognition of infectious agents by DC has been demonstrated to have a relationship with ITP. For example, TLR-4 that binds LPS, a Gram-negative bacterial endotoxin, was shown to be related to LPS-induced ITP [86,87]. TLR-7 was also shown to be related to B cell proliferation and therefore to anti-platelet antibody production in vitro via BAFF (B cell-activating factor of TNF family) production [88]. In addition, the so-called cryptic epitopes, normally not exposed in a self-structure, but which might become expressed and recognized by the immune system in special situations, such as an infection, have been described. The processing of these self-structures in the APCs produces a certain number of new epitopes, a process called epitope spreading, that can perpetuate and amplify the inflammatory response [86]. The T cell clones that interact with these cryptic epitopes might escape negative selection in the thymus when these self-determinants are present under a certain threshold concentration. For example, in ITP, platelets coated with anti-GPIIbIIIA are internalized and together with other platelet glycoproteins, such as GPIb/IX, are degraded inside the APCs and lead to the formation of a multitude of cryptic epitopes. Consequently, the APCs presenting these novel peptides, along with high levels of co-stimulatory molecules, determine the activation of naive T cells against the above-mentioned self-structures; in other words, the epitope spreading, mediated by APCs, has an important role in the generation and perpetuation of ITP [86]. 

DCs are also involved in inducing tolerance, a function provided by indoleamine 2,3-dioxygenase (IDO), an enzyme involved in the conversion of tryptophan (an important amino acid for T cell survival) into pro-apoptotic metabolites, such as kynurenine [80]. These DCs also determine the transformation of naive T cells into Tregs, which, in turn, maintain the tolerogenic state by increasing IDO expression by allowing the ligation of CTLA-4, which they constitutively express to CD80 and CD86 [80]. CTLA-4 is a transmembrane molecule present on activated T cells that, after binding to the B7.1/B7.2 heterodimer, will initiate a negative signal and therefore the inhibition of activated T cells, contributing to peripheral tolerance. It has been shown that in ITP, mature monocyte-derived DCs express lower levels of IDO than the controls [89,90]. This reduced level of IDO expression leads to a decreased ability of Th conversion into Tregs and to an impaired function of Tregs, which are unable to inhibit T cell proliferation or to secrete IL-10 [89]. A special subset of DC, which are called plasmacytoid DC (pDC) and are specialized in type I interferon (IFNα and IFNβ) production [91,92], have lower levels in patients with either primary or secondary ITP [70]. Macrophages contribute, besides DCs, to platelet destruction through endocytosis and in antigen presentation followed by stimulation of antibody production against platelets [80]. There are reports describing platelets in the cytoplasm of the macrophages localized in the red pulp and marginal zone of the spleen [93]. It was proven in vitro that splenic macrophages can spontaneously stimulate anti-GPIIb/IIIa-specific T cells [94]. This is followed by antibody production by B cells, antibody coating of the platelets, and uptake of the coated platelets by macrophages using their FcγRI receptors [94]. In ITP, the monocytes were shown to display a proinflammatory profile characterized by an increased expression of FcγRI and an increased FcγRIIa/FcγRIIb ratio, associated with a higher ability of phagocytosis. This profile would be reversed during therapy with dexamethasone [95]. Another noticed feature of macrophages in ITP is that they have higher levels of CD86 and HLA-DR expression compared to controls [96]. Regarding NK cells, some studies indicated that their number is increased in patients with more severe ITP disease, but the number was not correlated with their functional features [97]. Later, it was demonstrated that, unlike CTLs, the NK cells were not involved in platelet lysis. Therefore, further studies are needed to assess the role of NK cells in the pathogenesis of ITP. The pathophysiological mechanisms are complex and intricated and are shown in Figure 1. 

## 3. ITP as a Thrombotic Disease

If ITP was initially considered a hemorrhagic disease, more recent studies suggest that ITP has an increased risk of thrombosis [24]. Aledort et al. initially reported that 10 (5%) out of 186 adults with chronic ITP had 18 thromboembolic or ischemic events [98]. A recent review of the literature suggests that around 30 reported cases of ITP with thrombotic events were identified and a total of 36 events were recognized in the last 10 years [99]. If the presence of thrombosis in patients with ITP was initially considered more anecdotal, being reported in the context of case reports, later studies showed that the occurrence of thrombosis in patients with ITP is not purely accidental and can be considered a clinical reality with an important impact on the management of the disease. Paradoxically, the risk of thrombosis is higher in ITP patients in comparison with the general population [100]. Langeberg et al. (2016) described in their paper (a meta-analysis of observational studies) the incidence rate of arterial and venous thromboembolism in patients with immune thrombocytopenia and the relative risk of arterial and venous thromboembolism in patients with ITP and comparable populations without ITP [101]. The results showed that the incidence of arterial thromboembolism per 100 patient-years among patients with ITP ranged from 1.0 to 2.8, and among the population without ITP it ranged from 0.7 to 1.8 [101]. The incidence of venous thromboembolism per 100 patient-years among patients with ITP ranged from 0.4 to 0.7, and among populations without ITP ranged from 0.1 to 0.4 [101]. Arterial thrombosis has been reported more frequently than venous thrombosis. Rashed et al. showed in their work that most of the patients (around 64%) developed arterial events while fewer developed venous thrombosis [99]. The main causes of thrombosis in ITP can be grouped into three categories: causes related to ITP as a disease, causes that are related to the treatment of ITP, and causes related to other comorbidities.

### 3.1. Thrombosis Related to ITP as a Disease

The pathophysiological mechanism for thrombosis in ITP remains unclear, but several mechanisms have been proposed. Severe bleeding episodes are relatively rare in some patients with ITP even if they have a low platelet count, which suggests that these patients may have a protective factor against bleeding [102].

Platelets in ITP patients are younger and more active compared to those in the general population, which increases the risk of thrombosis [103,104]. The presence of young, large, hyperactive platelets is typical for chronic thrombocytopenia. 

The key role in increasing the risk of thrombosis is attributed to the formation of platelet microparticles (PMPs). Starting from the basic idea that ITP is an autoimmune disease, it can be considered that the appearance of autoantibodies determines the fragmentation of platelets with the formation of PMPs [105]. PMPs are derived from platelet membranes that are less than 0.5 microns and cannot be detected in routine platelet counting. They are described in association with platelet activation [106]. ITP patients have increased amounts of platelet and red cell microparticles [107]. Platelet microparticles are involved in thrombosis production through an increased procoagulant activity due to their higher content of large von Willebrand factor multimers that play an important role in platelet aggregation and coagulation [108]. Moreover, PMPs are potentially thrombogenic due to the high surface expression of phosphatidylserine [109]. In recent years, the phenotypic and structural analysis of PMPs has received the attention of clinicians, being considered the most common microparticles in the body and considered to have a role in the context of cardiovascular, infectious, or neoplastic diseases [110]. Recent studies have shown that PMPs promote blood coagulation, being involved in thrombin generation through activation of extrinsic and intrinsic pathways via phosphatidylserine exposure [111]. The membrane proteins of PMPs, having an anion phospholipid surface, can increase the catalytic activity of tissue factors exacerbating blood clotting formation. The tissue factor (TF) that forms the TF:VIIa complex activates factor X and factor IX to initiate blood coagulation [112]. The presence of the TF on PMPs increases their procoagulant activity [113]. In a recent article, Wang et al. showed that the level of PMPs is increased in patients with pulmonary thromboembolism, being considered one of its mechanisms of production and can also be considered a marker of the evolution of thromboembolic disease [114]. 

Another mechanism that was reported to be involved in thrombosis is the presence of antiphospholipid bodies (APA) in patients with chronic ITP. Antiphospholipid bodies have a strong acquired risk for arterial and venous thrombosis, being the most common form of acquired thrombophilia. Some publications have shown the coexistence of APA and ITP [115,116]. According to the current guidelines, when APA are detected in an ITP patient without a history of thrombosis or obstetric complications, this finding will not change the diagnosis of primary ITP [117,118]. 

Endothelial activation is one mechanism that can explain the different grades of bleeding even if there are no important differences between the platelets count. Endothelial activation is involved in coronary disease, diabetes, hypertension, and cerebrovascular disease, and is being investigated for its role in thrombosis in ITP patients. Garabeta et al. (2020), measured the markers of endothelial activation including intercellular adhesion molecule-1 (ICAM-1), V-CAM, and thrombomodulin in 21 ITP patients, as well as E-selectin in 18 ITP patients. Higher levels of ICAM-1, thrombomodulin, and H3Cit-DNA were found in ITP patients compared with controls. This study showed that ITP patients have increased endothelial activation which may contribute to the intrinsic hypercoagulable state of ITP [119]. In addition, Efat et al. showed that there was a highly statistically significant difference between case and control regarding the mean level of VWF-Ag and V-CAM. These markers could serve as biomarkers for endothelial alterations and should be investigated as a predictor of thrombocytopenic bleeding [120]. 

The concept of rebalanced hemostasis was extended to ITP which tried to explain another mechanism of thrombosis. Kim and al. (2015) postulated that the anti-thrombotic mechanism is balanced by the pro-thrombotic phenotype through an elevation of their plasma vWF antigen levels and hemostatic changes that promote thrombosis. Measuring the vWF antigen levels and performing TEG analysis can be a decisive factor for the treatment strategy in ITP patients [121].

Complement activation can increase the risk of thrombosis in ITP patients. There is a close interaction between the complement and hemostatic system that provides the role of complement in hematologic disorders. Hamad et al. (2008) reported that there is an interplay between complement and platelets: complement activates platelets and thrombin-activated platelets, in turn, activate the complement cascade [122]. Researchers showed that platelet aggregation and serotonin secretion were increased by the combination of C3 and the terminal complement complex [123]. On the other hand, there is crosstalk between complement and coagulation. The complement system activates coagulation cascade via multiple pathways. In vitro data showed that MASP-2 participated in the activation of thrombin and the subsequent generation of fibrin [124]. In addition, the complement activation products—C5a and the terminal complement complex—triggered tissue factor expression and activated endothelial cells, which resulted in the activation of the extrinsic coagulation pathway [123]. 

Acquired deficiency of natural anticoagulants in ITP patients can be explained by complement activation, through the terminal complement complex that inhibits natural anticoagulant, which results in an increased risk of thrombosis [123]. Tan et al. (2021) studied pulmonary thromboembolism in immune thrombocytopenia and showed that the level of antithrombin was deceased and the level of active protein C was increased, suggesting a link between these markers and thrombosis in this clinical context [108]. 

### 3.2. Thrombosis Related to ITP Treatment

According to actual guidelines, corticosteroids, intravenous immunoglobulin, and anti-RhD immune globulin are typical first-line and rescue treatments [125]. In about 70% of adult cases, ITP becomes persistent (lasting >3 months) or chronic (lasting >12 months) despite first-line treatment [125,126]. Second-line therapy options are currently represented by the thrombopoietin receptor agonists (Eltrombopag, Romiplostim), rituximab, and splenectomy [2]. The most recent American Society of Hematology ITP clinical guidelines recommend thrombopoietin receptor agonists (TPO-RA) rather than rituximab and rituximab over splenectomy [2]. New therapies that have appeared in recent years or are still under investigation in ITP involve new TPO-RA (Avatrombopag, Lusutrombopag), tyrosine kinase inhibitors (Fostamatinib, Rilzabrutinib), and protease inhibitors (Bortezomib) [125]. Salvage therapies involve therapeutic agents that are typically used after failure of multiple standard second-line treatment options, these include immunosuppressants (cyclophosphamide, cyclosporine, mycophenolate mofetil, azathioprine, vinca alkaloids) or danazol and dapsone [127]. 

The most studied and reported therapeutical factors of thrombosis in ITP were corticosteroids, immunoglobulins, TPO-RA, and splenectomy. One explanation is that these drugs can rapidly increase the platelet count and cause an increased risk of thrombosis [128]. Girolami et al. suggested in their work that prednisone-treated patients with immune thrombocytopenia can be considered potentially thrombophilic [129]. The mechanisms by which corticosteroids contribute to the thrombophilic status are characterized by the elevated factor VIII level, decreased fibrinolysis, and abnormal von Willebrand factor multimers [129]. Other authors showed that, although intravenous immunoglobulin is an effective treatment, thrombotic complications can occur [130]. Intravenous immunoglobulins use is related to blood viscosity increases and secondary hypercoagulable states. In addition, some immunoglobulin products contain factor XI or have direct effects on vascular endothelial cells and related vasospasm [131,132].

The efficacy of TPO-RA in ITP patients is attributed to their ability to increase platelet production and promote megakaryocyte survival. Increasing the platelet count above the normal target is a potential side effect and might increase the risk of thrombosis in patients treated with TPO-RA. In the context of ITP, megakaryocyte activation itself increases the risk of thrombosis [99]. The two TPO-RA, romiplostim and eltrombopag, are currently used as second-line therapies to increase the platelet count and prevent hemorrhagic events. Both bind to the thrombopoietin (TPO) receptor causing the activation of the JAK2/STAT5 pathway that increases the megakaryocyte progenitor proliferation and platelet production [133]. Regarding the mechanism of action, there are some differences between romiplostim and eltrombopag. Eltrombopag is a small molecule that binds at a transmembrane site of the thrombopoietin receptor and stimulates megakaryocyte precursors and differentiation [134]. Romiplostim binds directly and competitively at the TPO binding site and mostly stimulates mature precursors. [133,135]. The annualized thrombosis rates in adults appear to be 2–3 times higher with TPO-RA treatment than in an ITP not treated with TPO-RA, and even higher if compared to non-ITP control populations [133,136]. In general, thrombotic events tended to occur in the first year of treatment and occurred with high, normal, or low platelet levels, and mostly in those with at least one thrombosis risk factor in both eltrombopag and romiplostim [136]. Kuter 2021, recommends that romiplostim use „should be carefully considered in patients with thrombotic risk factors” [137]. A recent study suggests that there is a pro-coagulant state in ITP patients treated with TPO-RA [138]. This pro-coagulant state involves increased plasminogen activator inhibitor-1 (PAI-1) levels, increased platelet apoptosis that causes a higher exposure of phosphatidylserine, and basal exposure of P-selectin in quiescent platelets that were significantly increased in TPO-RA-treated patients compared to pre-treatment levels or to untreated patients [138,139]. 

Even if splenectomy has been used as a second-line treatment for ITP for 100 years, the rate of splenectomy for ITP is falling worldwide [140]. Splenectomy, being a surgical intervention, increases the risk of sepsis and thrombosis. Regarding this clinical aspect, Boyle et al. conclude that ITP patients, post-splenectomy, have an increased risk of abdominal thrombosis, deep venous thrombosis, pulmonary thromboembolism, and sepsis [141] Doobaree et al. (2016), in their meta-analysis, showed that there is an increased risk of thrombosis among ITP individuals and a higher risk of venous thrombosis after splenectomy [142]. They found that individuals who underwent splenectomy for ITP had a low prevalence of a history of thrombotic events prior to their procedure. The same study showed that splenectomised ITP individuals were also found to have a higher ‘annualised’ cumulative risk for venous thrombotic events than for arterial thrombotic events [142]. Vascular events after splenectomy are multifactorial, probably resulting from some combination of platelet activation, hypercoagulability, disturbance, activation of the endothelium, and altered lipid profiles [143]. Thromboelastography confirmed that the hyper-coagulant state is evident post-splenectomy [144]. This hypercoagulability can be associated with increased postoperative levels of platelets, fibrinogen, PAI-1, plasminogen activators, and activated partial thromboplastin time [145]. The hemodynamic effect of splenic vein ligation is a decrease in portal blood flow that increases the risk of thrombosis [146].

### 3.3. Thrombosis Related with Comorbidities in ITP Patients

There is a paucity of epidemiological data on the risk of comorbidities in adults with persistent or chronic immune thrombocytopenic purpura. Other risk factors such as diabetes, hypertension, cardiovascular disease, old age, and obesity may contribute to the occurrence of thrombotic events alone or in association with additional risk factors related to ITP treatment or ITP as disease. A French nationwide cohort study published in 2021 [147] that included 10,039 patients assessed the risk factor for venous thrombosis and arterial thrombosis for patients with primary immune thrombocytopenic purpura. The results of this study showed that the risk for venous thrombosis during hospitalization was observed in patients with a history of thrombosis, history of cancer, old age, and splenectomy. A higher risk of hospitalization for arterial thrombosis was observed in patients with older age, male sex, a history of cardiovascular disease, and splenectomy [147]. Another epidemiological report showed that smoking, hypertension, male gender, a history of thrombosis, and atrial fibrillation were significantly associated with the occurrence of thrombosis in ITP patients [148].

The main mechanisms involved to increase the risk of thrombosis in ITP are shown in Figure 2.

### 3.4. The Management of Thrombotic Events in ITP

Clinicians facing patients with ITP requiring treatment should be aware that there is a risk of thrombosis and should pay special attention and follow up. In patients with additional risk factors for thrombosis such as smoking, obesity, arterial hypertension, diabetes mellitus, atrial fibrillation, hypercholesterolemia, coronary disease, prolonged treatment with corticosteroids, or immobilization, efforts should be made to improve modifiable personal risk factors [136]. At the same time, physicians should be aware that all treatments for ITP patients have side effects, and their use is justified by an increased risk of bleeding for the patient and not only by correcting platelet counts—in order to produce a favorable benefit–risk ratio [149]. Avoiding a prolonged course of corticosteroids can be a measure to decrease the risk of thrombosis. According to this, in adults with newly diagnosed ITP, the ASH guideline panel recommends against a prolonged course (>6 weeks including treatment and taper) of prednisone and in favor of a short course (≤6 weeks) [2]. Prophylaxis for thrombosis should be performed in ITP patients in situations such as prolonged immobilization and surgery [136]. ITP patients often have comorbidities such as coronary disease, atrial fibrillation, and venous thromboembolism that require antiplatelet or anticoagulant therapy. No direct evidence exists to guide hematologists for the management of antithrombotic therapy in patients with ITP. Antithrombotic therapy in ITP is challenging and there are no recommendations for minimum platelet count thresholds. A study from 2016 showed a recommendation for a minimum platelet count of 50 × 10^9^/L for therapeutic anticoagulation in patients with chemotherapy-induced thrombocytopenia [150]. Another study published in 2018 that investigated ITP specialists and hematologists/oncologists revealed some interesting ideas regarding the antiplatelet and anticoagulant treatments in ITP patients. This study showed that there is a wide variation among practicing hematologists in the minimum platelet count recommended for patients with ITP who require antithrombotic therapy, and a platelet count of 50 × 10^9^/L was the most recommended minimum platelet count [151]. Some authors recommend that in patients with ITP and thrombosis, anticoagulation should be continued despite low platelet counts along with adequate treatment for ITP unless there is a very low platelet count (<20 × 10^9^/mL) associated with a significant risk of bleeding [152,153]. In ITP patients not responsive to corticosteroids, intravenous immunoglobulin or TPO-RA should be strictly used to improve platelet count to a level sufficient to safely administer antithrombotic therapy [136]. Management of thrombotic events in ITP should be individualized, also taking into account personal risk factors, risk of bleeding, and the severity of thrombosis.

## 4. Conclusions

Primary immune thrombocytopenic purpura might be considered a bleeding disorder with a high risk of thrombosis. The clinical manifestation of ITP varies between patients and is related to the pathophysiological mechanism of thrombocytopenia. The autoimmune response in ITP involves both the innate and adaptive immune systems, comprising both humoral and cell-mediated immune responses. Due to a deficiency in regulator mechanisms involving Tregs, IL-10-producing B cells, and tolerogenic DCs, this autoimmune mechanism is not counteracted. In contrast, the peripheral destruction of platelets cannot be overcome by an increase in platelet production since megakaryocytes are the target of both autoantibodies and CTLs and because the TPO has insufficient levels. Thrombosis in ITP is related to the pathophysiology of the disease (young hyperactive platelets, PMPs, rebalanced hemostasis, complement activation, endothelial activation, antiphospholipid antibodies, inhibition of natural anticoagulants), ITP treatment, and other comorbidities that altogether contribute to the occurrence of thrombosis. Physicians need to be vigilant in the early diagnosis of thrombotic events and then institute proper treatment (antiaggregant, anticoagulant) along with ITP-targeted therapy.

## Figures and Tables

**Figure 1 medicina-58-00211-f001:**
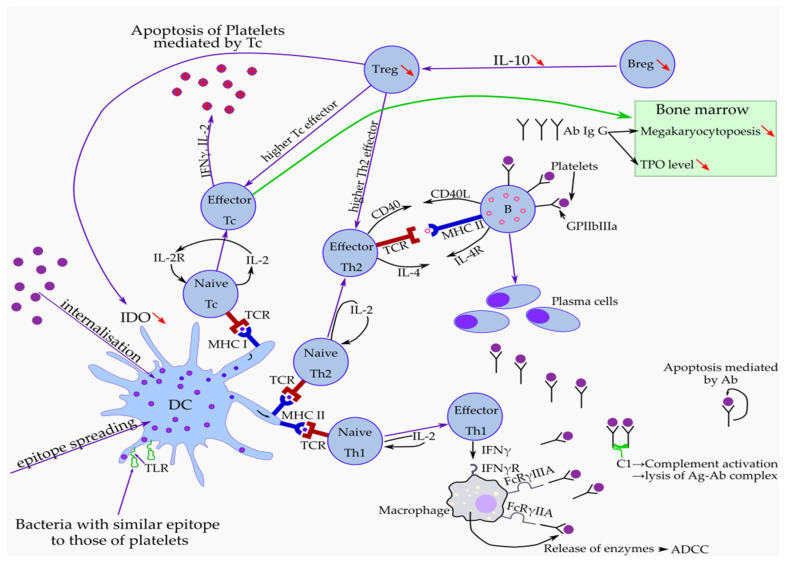
Pathophysiological mechanisms that produce thrombocytopenia and hemorrhagic events in ITP. DC—dendritic cells, IDO—indoleamine 2,3-dioxygenase, Tc—T cytotoxic, Treg—regulatory T cell, Breg—regulatory B cell, TPO—thrombopoietin, Th1—T helper 1, Th2—T helper 2, Ab—autoantibodies, Ag-Ab—antigen, antibody complex, TCR—T cell receptor, MHC I—Major histocompatibility complex class I, MHC II—major histocompatibility complex class II, IFNγ—interferon gamma, IL-2—interleukin 2, IL-10—interleukin 10, IL-4—interleukin 4, FcRγ IIA, IIIA—receptor for Fc region of Ig G, Ig G—immunoglobulin G, C1—complement component C1, ADCC—antibody-dependent cellular cytotoxicity.

**Figure 2 medicina-58-00211-f002:**
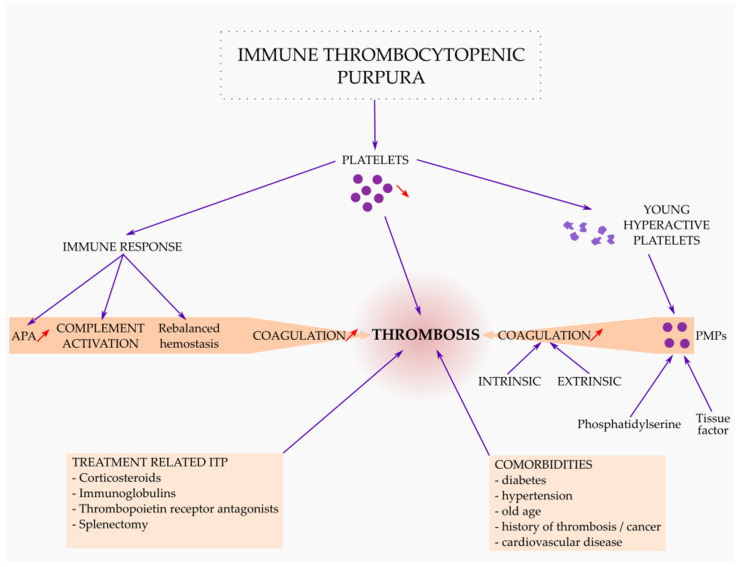
The main mechanisms involved to increase the risk of thrombosis in immune thrombocytopenic purpura (ITP). Thrombosis in ITP is related to the pathophysiology of the disease (young hyperactive platelets, PMPs—platelet microparticles, rebalanced hemostasis, complement activation, endothelial activation, APA—antiphospholipid antibodies, inhibition of natural anticoagulants), ITP treatment and other comorbidities.

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
