# Peer review of "Immune Thrombocytopenic Purpura as a Hemorrhagic Versus Thrombotic Disease: An Updated Insight into Pathophysiological Mechanisms"

_medicina, 2022, doi:10.3390/medicina58020211_

Round 1

Reviewer 1 Report

Thank you for allowing me to review this manuscript entitled “Immune Thrombocytopenic Purpura as a Haemorrhagic Versus Thrombotic Disease.” This is a review article challenging the understanding that ITP is classically a bleeding disorder, providing evidence that these patients often have a risk of thrombosis. Overall, the article is comprehensive, but it is very apparent that the authors are not native English speakers, and they should ask an English-speaking proofreader to fix grammatical and syntax errors. These specific areas require revision:

  1. Section 1:
    1. Page 2, line 72: “newly IT” is a typo; should be ITP
    2. Page 2, line 76-77: “All this evidence supports the idea that these phases are not homogenous and can have good or poor patients’ outcomes.” Please move this sentence to the end of the paragraph (line 81), as this summarizes the paragraph, and provide additional references of the molecular and clinical studies needed to make this claim. Also change “patients’ outcomes” to “clinical outcomes.”
  2. Section 2:
    1. Section 2.1 can be shortened substantially or removed entirely as most of this information is not directly relevant to the rest of the article, and section 2.2 is already borderline overly detailed in the explanation of the immune response in ITP
  3. Section 3:
    1. Page 9, line 425: just use the abbreviation PMP, no need to spell out “platelet microparticles” again after it has already been defined
    2. Page 9, line 442: same thing as above; use only TF since “tissue factor” has already been defined
    3. Page 9, line 447: AF is not the typical English abbreviation for antiphospholipid antibodies; use APA instead (and change AF to APA throughout the rest of the article)
    4. Page 9, line 450: “According to ITP guidelines” – who is ITP? Please review and clarify
    5. Double check all abbreviations used throughout this section and the rest of the article, as some are inconsistent and/or undefined (e.g. VCAM-1 vs V-CAM)
    6. It would be helpful to add a brief section citing any recent publications that discuss the treatment of thrombotic events in patients with ITP (i.e. specific indications/contraindications of anticoagulants or other treatments in the setting of ITP).

Reviewer 2 Report

1. This paper would helped us understand more about the various immunological mechanisms of ITP and its comprehensive cause of tendency to thrombosis. Although clinical reviews of thrombosis in ITP have appeared in recent years (like British Journal of Haematology, 2021 194(5), 822-834), I think a detailed review of the latest pathophysiology is attractive in this paper. Why don’t you emphasize that thing in the title, abstract, or background?

2. In the thrombosis related to ITP treatment, whether or not there is an association with thrombopoietin receptor agonists has been widely reported and is of interest to clinicians. Why don't you add a little more review on that part?

3. There is also an metanalysis that showed increased risk of thrombosis among ITP individuals and a higher risk venous thrombosis after splenectomy (European Journal of Haematology. 2016;97(4):321-330).

4. In the part of thrombosis related with comorbidities in ITP patients, there is another epidemiological report like Annals of hematology. 2020;99(1):49-55.

Round 2

Reviewer 1 Report

Thanks for the edits and additions, the article is much better now.